# Responses of *HSP70* Gene to *Vibrio parahaemolyticus* Infection and Thermal Stress and Its Transcriptional Regulation Analysis in *Haliotis diversicolor*

**DOI:** 10.3390/molecules24010162

**Published:** 2019-01-03

**Authors:** Zhiqiang Fang, Yulong Sun, Xin Zhang, Guodong Wang, Yuting Li, Yilei Wang, Ziping Zhang

**Affiliations:** 1Key Laboratory of Healthy Mariculture for the East China Sea, Ministry of Agriculture, Fisheries College, Jimei University, Xiamen 361021, China; zqfang0721@163.com (Z.F.); gdwang@jmu.edu.cn (G.W.); 13123395315@163.com (Y.L.); 2College of Animal Science, Fujian Agriculture and Forestry University, Fuzhou 350002, China; ylsun1@126.com (Y.S.); zhangxinexe@126.com (X.Z.)

**Keywords:** *HSP70*, thermal, *Vibrio parahaemolyticus*, transcriptional regulation, *Haliotis diversicolor*

## Abstract

Heat-shock protein 70 (HSP70) is a molecular chaperone that plays critical roles in cell protein folding and metabolism, which helps to protect cells from unfavorable environmental stress. *Haliotis diversicolor* is one of the most important economic breeding species in the coastal provinces of south China. To date, the expression and transcriptional regulation of *HSP70* in *Haliotis diversicolor* (*HdHSP70*) has not been well characterized. In this study, the expression levels of *HdHSP70* gene in different tissues and different stress conditions were detected. The results showed that the *HdHSP70* gene was ubiquitously expressed in sampled tissues and was the highest in hepatopancreas, followed by hemocytes. In hepatopancreas and hemocytes, the *HdHSP70* gene was significantly up-regulated by *Vibrio parahaemolyticus* infection, thermal stress, and combined stress (*Vibrio parahaemolyticus* infection and thermal stress combination), indicating that *Hd*HSP70 is involved in the stress response and the regulation of innate immunity. Furthermore, a 2383 bp of 5′-flanking region sequence of the *HdHSP70* gene was cloned, and it contains a presumed core promoter region, a CpG island, a (TG)_39_ simple sequence repeat (SSR), and many potential transcription factor binding sites. The activity of *HdHSP70* promoter was evaluated by driving the expression of luciferase gene in HEK293FT cells. A series of experimental results indicated that the core promoter region is located between −189 bp and +46 bp, and high-temperature stress can increase the activity of *HdHSP70* promoter. Sequence-consecutive deletions of the luciferase reporter gene in HEK293FT cells revealed two possible promoter activity regions. To further identify the binding site of the key transcription factor in the two regions, two expression vectors with site-directed mutation were constructed. The results showed that the transcriptional activity of NF-1 site-directed mutation was significantly increased (*p* < 0.05), whereas the transcriptional activity of NF-κB site-directed mutation was significantly reduced. These results suggest that NF-1 and NF-κB may be two important transcription factors that regulate the expression of *HdHSP70* gene.

## 1. Introduction

Small abalone *Haliotis diversicolor* belongs to Mollusca, Gastropoda, Prosobranchia, Archaeogastropoda, Haliotidae and *Haliotis*. Due to its high nutritional and medicinal values, *H. diversicolor* has become one of the most important economic breeding species in the coastal provinces of south China. However, the deterioration of the living environment, such as high temperature, hypoxia and infection of pathogenic bacteria, has led to a sharp decline in the resources and aquaculture of *H. diversicolor* [1,2]. Temperature and pathogenic bacteria are the key factors affecting the growth and health of the abalone [3,4]. The optimum growth temperature of the abalone is 22–28 °C. In a study it was shown that, when the water temperature elevated from 28 °C to 32 °C, the phenol oxidase activity and phagocytic activities were negatively affected in the animals, indicating the importance of the environmental factors, mainly temperature, on the innate immune system regulation of the abalones [5]. Similarly, *Vibrio parahaemolyticus* has been described as the major pathogen that significantly affects the culture of *H. diversicolor* by invading the innate immune system of the cultured animals [5,6]. Therefore, understanding the mechanism behind the adverse effects of environmental factors, mainly elevated temperature, and pathogen invasion, on the innate immune system of abalone is of high importance.

The innate immune system is the body’s first line in which the relevant cells non-specifically recognize and act on the pathogen to protect against pathogen infection [7]. Heat-shock proteins (HSPs) are a group of highly conserved chaperone proteins expressed by the cell that respond to unfavorable environmental change [8]. HSPs have been found in almost all organisms, from bacteria to humans. Various stimuli including heat stress damage can induce the synthesis of HSPs, which increase the adaptability of the organism to the environmental stresses [9]. The HSPs buffer this environmental variation and are therefore important factors for the maintenance of homeostasis across environmental regimes [10]. HSPs are also a potentially important modulator of immune responses against many bacterial infections [11]. Therefore, it is necessary to understand the function and regulation mechanism of heat-shock protein. According to the sequence similarity and molecular size, six major HSP families have been identified, namely HSP110, HSP90, HSP70, HSP60, HSP40, and the small HSPs [12].

In the HSP families, HSP70 has been widely studied as a biomarker [13]. The amino acid sequence homology of HSP70 family is highly conserved [14], and their molecular weights are around 70 KDa. HSP70 family can be roughly divided into two types: structural HSP70 (heat-shock cognate protein 70, HSC70), and inducible HSP70 (heat-shock inducible protein 70, HSP70) [15,16]. HSC70 was expressed in all tested cells, but usually there was no significant change in expression levels under stress conditions [17]. HSP70 was generally expressed in small amounts in normal cells, but its expression level rapidly increased under stress conditions [18,19]. HSP70 has the function of resisting oxidation, participating in cellular immunity, and enhancing cell stress tolerance [20]. HSP70 and HSC70 are not only expressed in mammals but also in non-mammals, such as fish [21] and mollusks [22,23]. Many studies have proved that HSP70 exhibits physiological and ecological importance in response to pathogen infection and environmental stress. For example, in fish, heat shock was the most effective stress stimuli to induce HSP70 response compared to other stressors including hypoxia and air exposure [24]. In mollusks, *HSP70* transcripts increased significantly after acute heat stress [25,26]. Up-regulation of HSP70 was observed after *V. parahaemolyticus* infection in adult bay scallops *Argopecten irradians* [27]. The expression of *HSP70* in the zebra mussel *Dreissena polymorpha* showed a time-dependent increase after lipopolysaccharide (LPS) stimulation [28]. However, so far, the expression of *HSP70* gene in *H. diversicolor* (named *HdHSP70*) has rarely been reported.

Gene expression in eukaryotes is the result of interaction between transcription factors and cis-acting elements. In addition, the promoter is an important response element that regulates gene expression and determines gene activity. Therefore, the promoter is always a research hotspot [29]. The promoter is located near the transcription start sites of structural gene and binds directly to RNA polymerase and its transcription factors to determine whether gene transcription is activated. The transcription factors of eukaryotes have specific DNA binding domain, transcriptional activation domain and regulatory domain, which bind to specific promoter elements on the DNA sequence under the action of the transcription activator [30]; therefore, it plays an important role in regulating the transcription of target genes [31,32,33]. Oda et al. demonstrated that the promoter of *Medaka HSP70* gene is an inducible promoter, which can induce the expression of foreign genes in vivo, in vitro, and during early embryo development of the transgenic *Medaka* [34]. The regulatory sequences of eukaryotic genes usually contain binding sites for multiple transcription factors and provide a basis for combinatorial interactions among different factors. Stephanou et al. found that transcriptional activator (STAT-3) enhances the promoter activity of HSP70 in HepG2 hepatoma cells, and observed an additive effect on the increase in HSP70 promoter activity when STAT-1 and HSF-1 were binding each other [35]. Chen et al. found that deletion of transcription factor binding sites rich in AT sequence enhances the luciferase activity of the HSP70 gene of the fly *Liriomyza sativae*, speculating the element inhibits transcription of the gene [36]. Sistonen et al. found that two members of the human HSF family, HSF1 and HSF2, can be used as transcriptional activators of heat-shock gene expression and synergistically induce HSP70 gene transcription [37]. However, the transcriptional regulation of *HdHSP70* in *H. diversicolor* has not been well characterized.

Regarding several immune-related genes of *H. diversicolor* and their activation under different environmental stress and bacterial infections, there have been relevant studies in our laboratory. For example, Zhang et al. reported that the mRNA expression levels in either gills or hemocytes of *SalakB, SaAkirin2* and *AbNF-κB* significantly up-regulated post thermal stress and the injection of *V. parahaemolyticus* [38]. Huang et al. have confirmed that the expression level of the heat-shock-responsive genes were up-regulated under thermal and hypoxia stresses [39]. Most of these experiments explored the expression of genes at less than 96 h in different environmental emergency situations. Recently, Sun et al. found that the mRNA expression of the PI3K-AKT was significantly down-regulated after the *V. parahaemolyticus* stimulation with environment stimulation (thermal, hypoxia, and thermal and hypoxia) in gills, hemocytes, and hepatopancreas after more than 96 h [40]. Therefore, this project aims to investigate (1) the expression of *HdHSP70* gene under *V. parahaemolyticus*, thermal stress, and their combined stress for more than 96 h, and (2) the promotor structure and transcriptional regulation of *HSP70* gene in *H. diversicolor*. These studies can provide new insights into the immune mechanisms of *H. diversicolor.*

## 2. Results

### 2.1. Tissue Expression of the HdHSP70 Gene

The expression level of *HdHSP70* in different tissues was performed by quantitative real-time polymerase chain reaction (qRT-PCR). *HdHSP70* was expressed in all examined tissues with the significantly higher expression level being in hepatopancreas (Hp) (*p* < 0.05), followed by hemocytes (He) (*p* < 0.05) (Figure 1). Based on the obtained results, Hp and He were set as experimental tissues in different stress groups.

### 2.2. Expression of the HdHSP70 Gene under Different Stresses

In the Hp, under normal condition, the expression level of *HdHSP70* was significantly higher (*p* < 0.05) in NE group (abalones were injected with *V. parahaemolyticus* after they had been maintained for 96 h at the normal temperature (25 °C) and then continually being maintained at the normal temperature) than that in NB group (abalones were injected with 0.9% NaCl after they had been maintained for 96 h at the normal temperature (25 °C) and then continually being maintained at the normal temperature) at 24 h, but NE group and NB group were not different at 4 h, 12 h, and 48 h. Under the thermal stress condition, the expression level of *HdHSP70* was significantly higher (*p* < 0.05) in HE group (abalones were injected with *V. parahaemolyticus* after they had been maintained for 96 h at the thermal condition (30 °C) and then continually being maintained at the thermal condition) than that in HB group (abalones were injected with 0.9% NaCl after they had been maintained for 96 h at the thermal condition (30 °C) and then continually being maintained at the thermal condition) at 4 h and 12 h, but HE group and HB group were not statistically different at 24 h and 48 h (Figure 2).

In the He, under normal condition, the expression level of *HdHSP70* was significantly higher (*p* < 0.05) in NE group than that in NB group at 48 h, but NE group and NB group were not different at 4 h, 12 h and 24 h. Under the thermal stress condition, the expression level of *HdHSP70* in HE group was significantly higher (*p* < 0.05) than that in HB group at 12 h, 24 h and 48 h, but HE group and HB group were not statistically different at 4 h (Figure 3).

### 2.3. The Sequence Analysis of the 5′-Upstream Regulatory Region of HdHSP70 Gene

The 2383 bp 5′-flanking sequence of *HdHSP70* gene was obtained by Tail-PCR and Genome Walker method (Figure 4). The prediction of online software BDGP (Berkeley Drosophila Genome Project) showed that the highest area of the *HdHSP70* score (score = 1) may be the location of the transcriptional starting site (TSS) of the 5′-regulatory region of the *HdHSP70* gene, which is defined as the position of the presumed core promoter, starting from –40 bp to +10 bp region, and a typical TATA (a sequence of DNA in the core promoter region of genes) box is located at –27 bp to –32 bp ahead of the TSS, but no canonical CAAT box or GC box was found. The potential transcription factor binding sites including NF-1, TBP, C/EBPalpha, OCT-1, GCN4 and NF-κB, etc. and 5 heat-shock elements (HSEs) were found by Alibaba2. In addition, analysis of 5′-flanking sequence revealed that one (TG)_39_ repeat was located between −962 bp and −1040 bp. CpG island prediction software analysis (GC (a cytosine nucleotide is followed by a guanine nucleotide in the linear sequence of bases along its 5′→3′ direction) Percent > 50.0, Obs./Exp. > 0.6, length > 100 bp) found a CpG island with a length of 228 bp located at −152 bp to +76 bp.

### 2.4. Activity Analysis of the Promoter Region of HdHSP70 Gene

To identify the promoter activity of the *HdHSP70* gene, the complete 2383 bp 5′-upstream region was inserted into the pEGFP-1 vector (pEGFP-HSP70) and used to drive expression of the EGFP (Enhanced Green Fluorescent Protein) gene in HEK293FT cells. Under fluorescence microscope, the green fluorescence signal could be detected in pEGFP-HSP70, but it was lower than positive control pEGFP-N1. Meanwhile, no green fluorescence activity was detected in pEGFP-1 (without promoter). The results indicated pEGFP-HSP70 can drive the expression of EGFP protein in HEK239FT cell line (Figure 5).

To identify the core promoter region of the *HdHSP70* gene, two constructed reporter plasmids were prepared and transfected into HEK293FT cells. The plasmid with one from −1974 to +46 containing the core promoter region was named pGL3-70-1 and the other one from −1974 to −189 lacking the core promoter region was named pGL3-70-1d. The result showed that the luciferase activity of pGL3-70-1 was significantly higher than pGL3-70-1d and negative control (pGL3-Basic, plasmid without inserts) (*p* < 0.01), indicating the core promoter region was located between −189 bp and +46 bp (Figure 6).

To investigate if the *HdHSP70* promoter-driven luciferase reporter gene is induced by heat shock, HEK293FT cells was exposed to high temperatures, ranging from 37 °C to 39 °C and also 42 °C for 40 min, we found a significant increase in the activities of luciferase at 39 °C and 42 °C compared to 37 °C (*p* < 0.05), with the highest activity being observed at 39 °C (Figure 7).

To identify important transcription factor binding sites in the *HdHSP70* promoter region, the continuous truncated promoter fragments of the *HdHSP70* gene by PCR were amplified and cloned into luciferase reporter vectors. These recombinant plasmids were named pGL3-70-1, pGL3-70-2, pGL3-70-3, pGL3-70-4, pGL3-70-5, pGL3-70-6, and pGL3-70-7 and were used to transfect into HEK293FT cells. The results showed that most of luciferase recombinant plasmids, from pGL3-70-1 to pGL3-70-7 had greater activity than the control group (pGL3-Basic, *p* < 0.05). The luciferase activity of pGL3-70-2 was significantly higher than other recombinant plasmids (*p* < 0.05), suggesting that important regulatory elements exist in the pGL3-70-1 to pGL3-70-3 regions of the *HdHSP70* gene promoter (Figure 8).

Based on the results obtained, we hope to find regulatory transcription factors from pGL3-70-1 to pGL3-70-2 and from pGL3-70-2 to pGL3-70-3, respectively. The distance between the two deletion fragments of pGL3-70-1 and pGL3-70-2 contains a sole NF-1 binding site (TGACATTTCA) in the promoter of *HdHSP70*. Thus, we hypothesized that NF-1 is an important transcriptional binding factor for *HdHSP70*. To test our hypothesis, the AC (an adenine nucleotide is followed by a cytosine nucleotide in the linear sequence of bases along its 5′→3′ direction) of the sequence of NF-1 binding site was mutated to CT (a cytosine nucleotide is followed by a thymine nucleotide in the linear sequence of bases along its 5′→3′ direction), and the sequence became TGCTATTTCA instead of TGACATTTCA, and the mutant recombinant plasmid was named pGL3-NF-mut. A recombinant plasmid containing wild type of NF-1 binding site was also constructed and named pGL3-NF-wt. The luciferase activity analysis showed that in pGL3-NF-mut, luciferase activity was significantly increased compared to the same in pGL3-NF-wt (*p* < 0.05) (Figure 9). Similarly, the distance between the two deletion fragments of pGL3-70-2 and pGL3-70-3 contains a sole NF-κB binding site in the promoter of *HdHSP70*. Thus, we hypothesized that NF-κB is an important transcriptional binding factor for *HdHSP70*. To test our hypothesis, a recombinant plasmid containing NF-κB binding site GCTTAAGAGGAATTT was constructed and named pGL3-NK-wt. A recombinant plasmid in which GCTT was deleted to achieve a sequence AAGAGGAATTT instead of GCTTAAGAGGAATTT was constructed and named pGL3-NK-mut. The luciferase activity analysis showed that in pGL3-NK-mut, luciferase activity was significantly decreased compared to the same in pGL3-NK-wt (*p* < 0.05) (Figure 10).

## 3. Discussion

A lot of evidence suggests that HSP70 is an activator of the innate immune system [41,42,43]. When the body is stimulated by unfavorable factors such as changes in temperature, hypoxia, and pathogenic infection [44], HSP70 can be expressed in a short time to increase the tolerance of the body to adverse factors. For example, Aleng et al. reported that non-lethal heat shock (NLHS) treatment helped Asian green mussel *Perna viridis* to tolerate pathogen infection [45]. Junprung et al. suggested that NLHS treatment enhanced the tolerance of *Penaeus vannamei* to *V. parahaemolyticus* infection [43]. However, the effect of HSP70 on *H. diversicolor* immune function has rarely been studied. Our results provide a relevant basis for *HdHSP70* to involve in the heat-induced/infective immune function of *H. diversicolor* and that *HdHSP70* expression is regulated by related transcription factors.

### 3.1. The Expression of HdHSP70 mRNA in Different Tissues and Different Stresses

The study of the HSP70 family has confirmed the importance of cell self-protection by controlling protein metabolism under stress conditions [24,46]. Therefore, studying the distribution of *HdHSP70* mRNA in different tissues and different stresses will help us understand its stress response mechanism. According to the results of qRT-PCR, the expression of *HdHSP70* gene in Hp and He was significantly different from other tissues, especially in Hp. It has been reported that Hp and He play an important role in the innate immune system in mollusks [47,48]. Therefore, Hp and He may be more sensitive to environmental stress than other tissues. Many studies have similar results, for example, the higher expression of the *HSC70* mRNA in HP and HE of *H. diversicolor* [49], and higher expression of *HSP70* mRNA in the blood of *Oncorhynchus mykiss* [50]. Norouzitallab et al. reported that the increase gradually expression of *HSP70* in the mild heat stress of *Artemia* [41]. Junprung et al. found that NLHS could induce significant up-regulation of *HSP70* gene in hemocytes of *P. vannamei* in 0–6 h [43].

Studies have confirmed that natural fluctuations in environmental temperature and other physical and chemical parameters can lead to the induction of cell stress responses in some mussels and marine snails, of which the HSP family plays an important role [51,52]. Tirard et al. have reported that hemocytes of the *Crassostrea virginica* exhibited a strong heat-shock response when undergoing a sharp rise in temperature, which characterized by enhanced synthesis of several proteins [53]. Furthermore, HSP levels are not only indicators of heat exposure. In winter, the expression level of *HSP70* mRNA by *Mytilus californianus* was higher than in summer [54]. Encomio and Chu observed that the total amount of *HSP70* in the *C. virginica* did not have a positive correlation with seasonal changes in temperature [55]. It has also been reported that *HSP70* increased in the tolerance of *P. vannamei* to acute hepato-pancreatic necrosis disease (AHPND) due to *V. parahaemolyticus* infection [43]. These different results indicate that the expression of *HSP70* is plastic, which may depend on the type of environmental stresses, intensity, and duration.

Our results showed that the relationship between *HSP70* mRNA levels and elevated temperatures and *V. parahaemolyticus* infection was similar to these studies. Regardless of the normal or thermal stress condition in the Hp and He, the expression level of *HdHSP70* in *V. parahaemolyticus* infection group (NE and HE group) was significantly higher than control group. In Hp, high expression of *HdHSP70* occurs in HE at 4 h and 12 h and NE appears at 24 h. In He, high expression of *HdHSP70* occurs in HE at 12 h, 24 h and 48 h and in NE for 48 h. Interestingly, in our experimental results, the high expression time of *HdHSP70* of HE group and NE group in the Hp was obviously earlier than that in the He. In Hp, the highest expression of *HdHSP70* appeared in HE at 12 h and in NE at 24 h; however, in He, the highest expression of *HdHSP70* appeared in HE at 24 h and NE at 48h. This is probably because HSP70 in the Hp responds to stress more preferentially than He. It is also possible that in different tissues, there are probably different immune regulatory mechanisms to face the body’s stress. Although these immunomodulatory mechanisms are not yet clear in *H. diversicolor* at present, the fact is apparent that the increase of *HSP70* mRNA expression after stress enables the cells, tissues, and even the whole organism to obtain higher and earlier anti-stress ability.

### 3.2. Analysis of the 5′-Flanking Sequence of the HSP70 Gene

CpG island is a fragment with high GC content, which is located at the 5′-end of the gene. It plays an important role in regulating gene expression [56,57]. The predicted CpG island is close to TSS of the *HdHSP70* gene that is different from the CpG island of the *HSC70* gene in *H. diversicolor*, which is far from TSS [49]. Studies have shown that typical CpG islands appear on or near the TSS of most genes in eukaryotes, so many studies have focused on the proximal CpG island of the promoter and found that it can initiate transcription [58]. Conversely, Sarda et al. found that the CpG island in the MethExp gene (away from the annotated TSS) has promoter-like characteristics and was involved in transcriptional regulation of genes [59]. Whether there is a difference of CpG island between *HdHSP70* and *HdHSC70* influences transcription level is worth exploring.

Microsatellite DNAs or simple sequence repeats (SSRs) are formed by tandem repeats of short core sequences [60], which has become one of the hottest topics in genetics and ecology in recent years. Many studies have shown that SSR repeats appear to be a key regulatory factor in gene expression and expression levels. It has been found that some AT-rich sequence elements in the promoter region contain transcription factor binding sites, which play a significant regulatory role [61,62]. Some reports have also demonstrated that SSRs of different sizes in eukaryotic promoters can cause differential gene expression [63,64]. Shimajiri et al. found that the promoter activity of the matrix metalloproteinase-9 gene was down-regulated by shortening the (CA)_21_ microsatellite sequence [65]. In addition, some genes can only be expressed in a specific number of replicates of SSR. For example, (GAA)_12_ in the *E. coli* lacZ gene promoter allows expression of the lacZ gene, while neither (GAA)_14–16_ nor (GAA)_5–11_ allow expression of this gene [66]. The 5′- regulatory region of *HSP70* gene contains a (TG)_39_ microsatellite, which may be used to study the relationship between the number of microsatellites and the regulation of *HdHSP70* gene expression.

The promoter region of the HSPs gene often contain the binding sequences of a series of trans-factors that induce the basic transcription, such as SP1, CAAT (a sequence of DNA in the core promoter region of genes) and TATA boxes, and with some HSEs in their 5′-upstream. Zhuang et al. cloned the 5′-flanking regions of the 538 bp and 305 bp lengths of the *Bombyx mori HSP70* genes and found that HSE existed in the upstream of the TATA-box sequence [67]. Kust et al. found that the 5′-regulatory region of the Drosophila *HSP70* promoter contains 4 and 8 HSEs sequences and the promoter activation required 24 h after heat shock for the constructs with eight HSEs, but those with four HSEs required 48 h [68]. Another study showed that the promoters of *HSP70* genes contain multiple copies of HSE located at variable distance 5′-end to the TATA box, and the proximal HSE (nearest to the TATA box) is the most important in transcriptional activation of the *HSP70* gene [69]. Tsutsumiishii et al. used different reporter constructs containing proximal HSE, distal HSE to identify HSE and found that proximal HSE is required for activation of the HSP70 promoter [70]. Interestingly, the distal HSE (farther away from the TATA box) in the *Xenopus* HSP70 gene was heat-induced in monkey COS cells [71]. However, the distal HSE of the *Drosophila* HSP23 gene was not heat-induced in same cells [72]. Apparently, the HSE positional relationship in the heat-inducible promoter is not absolute, but the presence of multiple HSE binding sites in most genes is significant. There are 5 HSEs in the 5′-regulatory region of *HdHSP70* (one HSE is closer to the TATA box and the other four are farther), and their number and distribution are similar to HSE in the *HSC70* gene of *H. diversicolor* [49]. Although there was no typical correlation between the numbers or position of HSE and the degree of heat-shock-induced expression, it is presumed that they play a regulatory role in the expression of *HdHSP70* gene.

### 3.3. Analysis of the Activity of the HdHSP70 Promoter

The promoter of the HSPs gene is an inducible promoter whose activity is significantly increased under high temperature or other stress conditions [72]. Li et al. studied the heat-induced expression of human *HSP70B*’ in vitro, and transfected the GFP heat-inducible expression vector pHSP70B’-GFP into human breast cancer MCF-7 cells [73], and Zhuang et al. transferred into BmN cells with the 305 bp *HSP70* promoter of *Bombyx mori* [67]. Their results showed that both luciferase activities increased significantly with increasing induction temperature, reflecting the activity of the HSP70 heat-shock promoter. In our experiment, the luciferase activity of *HdHSP70* was significantly increased in HEK293FT cells exposed to high-temperature stress, and the activity was highest at 39 °C, indicating that high-temperature stress will increase the activity of *HdHSP70* promoter. However, similar to Li et al. results, too high temperature (42 °C) will lead to the decrease of the promoter activity [73].

In the process of transcriptional regulation of genes, promoters regulate the expression of genes under the synergistic action of certain positive or negative regulatory transcription factors [49]. Our experimental results showed the activity of the deletion fragment pGL3-70-2 was significantly higher than other deleted fragments. After mutation in the transcription factor binding site NF-1 between pGL3-70-1 and pGL3-70-2, *HdHSP70* promoter activity was inhibited. On the other hand, mutation in the transcription factor binding site NF-κB between pGL3-70-2 and pGL3-70-3, *HdHSP70* promoter activity was enhanced.

Nuclear factor-1 (NF-1) is a transcription factor that plays an important role in the process of gene expression. It can regulate the expression of various genes by combining with the specific identification sequence of the gene promoter [74]. For example, some studies have shown that NF-1 plays an important role in the transcription of type I collagen a1 chain gene and a2 chain gene. Without NF-1, transcription cannot proceed normally [75,76,77]. The transcription factor NF-1 plays a significant up-regulation role in the transcriptional expression of the human α1 collagen gene [74]. However, some studies have also found that NF-1 can up-regulate the proliferative activity of cells by inhibiting the expression of the *P21* gene in CDK (cyclin-dependent kinase) inhibitory proteins [78]. Similarly, Borengassar et al. studied the regulation of NF-1 on the expression of Sulfotransferase isoform 1A1 (*SULT1A1*) gene in breast cancer in vitro, and found that NF1A, NF1B and NF1C can down-regulate the expression of *SULT1A1* [79]. In our experiment, although the NF-1 transcription factor has a certain inhibitory effect on the *HdHSP70* gene, its regulation mechanism is still unclear and needs further investigation.

Nuclear factor kappa B (NF-κB) is a highly conserved multifunctional transcription factor that plays an important role in immune and inflammatory responses [80]. Its functions have been extensively studied in the regulation of immune-related genes in mammals and arthropods [81]. In mammals, NF-κB can regulate the expression of cytokines, cell adhesion molecules, stress response protein and other genes [82]. In the immune response of invertebrates, such as *Drosophila*, activation of NF-κB transcription factor can induce the expression of related immune genes [83]. Upon activation, NF-κB can regulate gene transcription by combining with transcription factor binding sites in promoters [84]. Ammirante et al. found that NF-κB activity could down-regulate the human *HSP90α* gene promoter, revealing the relationship between NF-κB and HSP in cell defense mechanism [85]. Similarly, Zhao et al. found that NF-κB inhibits the activity of the *HSC70* promoter in *Litopenaeus vannamei* [86]. However, Wilhide et al. found that the expression of *HSP70.3* and *HSP70.1* genes was significantly up-regulated by NF-κB when exploring the protective effect of coronary artery occlusion on the heart [87]. Zhang et al. found that NF-κB positively regulates the expression of molt-inhibiting hormone (*MIH*) gene in *Scylla paramamosain* [88]. Similar results were obtained in our experiments: NF-κB has a positive regulatory effect on *HdHSP70* promoter activity. However, the specific regulation mechanism of NF-κB on *HdHSP70* still needs further exploration.

## 4. Materials and Methods

### 4.1. Animals and Preparation of Samples

Adult small abalones (body length 6.00 ± 0.50 cm, weight 15.7 ± 2.48 g) were purchased from the Peiyang abalone farm (Xiamen, China). These abalones were fed with sea tangle once a day and maintained in recycling systems with sand-filtered seawater at temperature of 25 °C and dissolved oxygen (DO) of 6.2 mg/L (as a normal condition, according to our previously published peer reviewed article) [38]. To accommodate experimental animals to the new environment, the abalones were kept under the normal condition for 7 days before the challenge experiments. The water temperature and DO were monitored continuously during the whole period of the experiment. In thermal stress experiment, the water temperature and DO were set at 30 °C and 6.2 mg/L respectively [39,40].

Small abalones were randomly divided into four groups (as a different condition, according to our previously published peer reviewed article [40]): (1) two groups of animals were maintained for 96 h at the normal temperature (25 °C). After 96 h, two groups were cultured under normal condition at 96 h: for the experiment group, all individuals were injected with 50 μL live *V. parahaemolyticus* (1.1 × 10^8^ cfu/mL in 0.9% NaCl) into the foot muscle, indicted as NE, and for the blank group, all individuals were injected with 50 μL 0.9% NaCl, indicted as NB; (2) two groups maintained for 96 h at increased temperature (30 °C) for thermal stress experiment, 30 °C was achieved by increasing from 25 °C with 1 °C per hour according to our previous study [39]. After 96 h, for the experiment group, all individuals were injected with 50 μL live *V. parahaemolyticus* (1.1 × 10^8^ cfu/mL in 0.9% NaCl) into the foot muscle, indicted as HE, and for blank group, the individuals were injected with 50 μL 0.9% NaCl, indicted as HB. Hp and He were collected at 4, 12, 24 and 48 h after injection of live *V. parahaemolyticus* or 0.9% NaCl. At least six abalone’s Hp and He were sampled at different time phases in all four groups. The details of these groups were summarized in Figure 11. Hemocytes (He) were isolated by centrifugation at 2000× *g* for 10 min at 4 °C from hemolymph that was collected by cutting off the foot quickly. Subsequently, hemocytes were immediately stored in liquid nitrogen for RNA isolation and quantitative real-time PCR (qRT-PCR). At the same time, to assess *HdHSP70* gene expression in different tissues, hepatopancreas (Hp), gills (Gi), mucous gland (Mg), digestive tract (Di), mantle (Ma), and muscle (Mu) were sampled and frozen into liquid nitrogen immediately for RNA extraction and qRT-PCR.

### 4.2. Isolation and Reverse Transcription of Total RNA

Total RNA was extracted from different tissues and from Hp and He of four groups using Total RNA Extraction Kit (Promega, Shanghai, China) according to the manufacturer’s protocol. Total RNA quality was assessed by agarose gel electrophoresis and Nano Drop ND-1000. The complementary DNA (cDNA) was synthesized in 20 μL reaction system including 3 μg total RNA (treated with DNase I), 2 μL random primers (10 mM), 4 μL 5× First-strand Buffer, 1 μL dNTP mix (10 mM), and 1 μL M-MLV reverse transcriptase (200 U/μL) (Promega, Shanghai, China). According to our previous data [38,40,49], the synthesized cDNA was diluted by 10-fold and 100-fold and stored at −20 °C until use.

### 4.3. Analysis of HdHSP70 Gene Expression After Stresses by qRT-PCR

qRT-PCR was carried out in LightCycler 480 Roche Realtime Thermal Cycler (Armonk, NY, USA) in accordance with the manual with a 20 μL reaction volume containing 9 μL of 1:100 diluted original cDNA, 10 μL of 10× SYBR Green Master Mix (Promega, Shanghai, China), and 1 μL of forward and reverse primer mix (10 μM each). The cycling conditions for *HdHSP70* were set as follow: 1 min at 95 °C, then followed by 40 cycles at 95 °C for 15 s, 60 °C for 1 min. The products were assessed by electrophoresis with 1% agarose gel. The housekeeping gene *β-actin* (Table 1) was used as endogenous gene, and its stability has been confirmed in our previous studies [36,37,38]. Melting curves were also plotted (60 °C–90 °C) to confirm that a single PCR product was amplified. The relative quantification (RQ) of gene expression was calculated using ΔΔCT (comparative threshold cycle) method (ΔCT = CT of target gene minus CT of the *β-actin* gene, ΔΔCT = ΔCT of any sample minus calibrator sample) and analyzed with SPSS version 20.0 (IBM, Basel, Switzerland) for One-way analysis of variance (One-way ANOVA). The *t*-test was used to determine the difference in the mean values among the treatments. The statistically significant difference was shown at *p* < 0.05.

### 4.4. Cloning and Bioinformatic Analyses of the 5′-Flanking Regions of HdHSP70 Gene

The 5′-upstream regulatory sequences of *HdHSP70* gene (GenBank accession number FJ812177.1) were isolated using the Tail-PCR and Universal Genome Walker Kit (TaKaRa, Dalian, ChinaTaKaRa, Dalian, China). The primer sequences used in this study are listed in Table 1. PCR products were purified, the possible objective DNA fragments were cloned into pMD19-T vector (TaKaRa, Dalian, China), then positive recombinant plasmids were sent to Sangon (Shanghai, China) for DNA sequencing. To predict the core promoter region and the transcriptional starting site, the online software NNPP (Rockville, MD, USA) (http://www.fruitfly.org/seq_tools/promoter.html) was used with the minimum promoter score of 0.8. The potential important transcription factor binding sites were analyzed by applying the online search software Alibaba2 (Heidelberg, Germany) (http://www.gene-regulation.com/pub/programs/alibaba2/index.html) databases. The CpG island was predicted by using the MethPrimer (San Francisco, CA, USA) with default parameters (http://www.urogene.org/cgi-bin/methprimer/methprimer.cgi).

### 4.5. HdHSP70 Promoter Activity in pEGFP-1 Vector

The pEGFP-1 is a promoterless EGFP vector that can be used to monitor transcription from different promoters and promoter/enhancer combinations inserted into the MCS (multiple cloning site) located upstream of the EGFP coding sequence. To identify the promoter activity of the *HdHSP70* gene, the promoter sequence of *HdHSP70* gene from −1974 to +46 in the 5′-flanking region relative to the transcription start site (+1) was amplified using primers (Table 1), which contained the respective Xho I and Kpn I restriction sites and protective bases. The target PCR product was purified and ligated into the pEGFP-1 vector, which was also cut with Xho I and Kpn I. The recombinant vector was defined as pEGFP-HSP70, which was screened and sequenced for correctness. The pEGFP-1 and pEGFP-N1 plasmids were used as negative and positive controls, respectively. The promoter activity of the 5′-flanking region was then tested by transfection of the recombinant plasmid pEGFP-HSP70 into HEK293FT cells. The fluorescent signal of pEGFP-HSP70, pEGFP-1 and pEGFP-N1 were observed at 24 h post transfection under an inverted fluorescence microscope.

### 4.6. Construction and Identification of Missing Fragment Expression Vector

The backbone of the pGL3-Basic Luciferase Reporter Vectors is designed for increased expression and contains a modified coding region for firefly luciferase, which lacks eukaryotic promoter and enhancer sequences. In addition, it has been optimized for monitoring transcriptional activity in transfected eukaryotic cells. To further identify the core promoter region of the *HdHSP70* gene, two constructed reporter plasmids with pGL3-70-1 from −1974 to +46 containing the core promoter region and the other pGL3-70-1d from −1974 to −189 lacking the core promoter region were prepared and transfected into HEK239FT cells. Meanwhile, the negative control pGL3-Basic was set up. The two promoters’ activities were detected by luciferase activities. To investigate whether the *HdHSP70* promoter-driven luciferase reporter gene is induced by heat shock, HEK293FT cells were exposed to high temperatures of 37 °C to 39 °C and 42 °C for 40 min and then their luciferase activities were detected. To analyze whether the 5′-upstream region of the *HdHSP70* gene has regulatory effects, seven consecutive deletions from the 5′-flanking regions were amplified by PCR. The correct seven promoter-fragment-constructs were cut with Kpn I and Xho I and cloned into pGL3-Basic reporter vector, which was also cut with Kpn I and Xho I. Finally, seven target plasmids (pGL3-70-1, pGL3-70-2, pGL3-70-3, pGL3-70-4, pGL3-70-5, pGL3-70-6 and pGL3-70-7) were extracted and purified using EndoFree Mini Plasmid KIT II (Tiangen, Beijing, China) and transfected into HEK293FT cells to test the luciferase activity of recombinant plasmids of different lengths.

### 4.7. Mutation of Transcription Factor Binding Sites

To investigate the function of cis-acting elements, the point mutation of the binding sites of specific transcription factors was used in this experiment. Firstly, the different fragments of 5′-flanking sequence of *HdHSP70* gene with significantly different luciferase activities were obtained from above experiments. Secondly, the different transcription factors in these different fragments were selected. Lastly, mutations were created at binding sites of different transcription factors. In this experiment, primers containing mutated bases and Kpn I and Xho I restriction sites located at each 5′-end were designed for overlapping extension PCR reactions. The PCR product was cloned into the pMD19-T vector, and then the fragment was digested with Kpn I/Xho I and subcloned into the promoterless vector pGL3-Basic for transfection of HEK293FT cells.

### 4.8. Transient Transfection and Activity Assays of the Luciferase Reporter Plasmids

HEK 293FT cells were routinely cultured in DMEM high glucose medium. When the well-grown cells grew to 80%, the recipient cells were inoculated into the 48-well culture plate at a density of 1–3 × 10^5^ cells/well. According to the manufacturer’s recommendations the luciferase assay was performed using a Dual-Glo luciferase assay system (Promega, Madison, WI, USA) with pRL-TK vector expressing Renilla luciferase regulated by the herpes simplex virus thymidine kinase promoter. The ratio of the target plasmid to the internal plasmid was 20:1, and the ratio of Lipofectamine 2000 to plasmid was 2.1:1. Transfection reagents and plasmids were mixed and co-transfected into the HEK293FT cell line. The pEGFP-N1 plasmid was employed as the positive controls. At 24 h post transfection, the cells were lysed by shaking in 60 μL of lysis buffer for 15 min at room temperature and then transferred to a 1.5 mL tube. After centrifugation, 5 μL of the clarified supernatant was collected into a new 1.5 mL tube. Firefly and Renilla luciferase activities were measured by the Dual-Luciferase^®^ Reporter Assay System (Promega, Madison, WI, USA) at the manufacturer’s protocol and chemiluminescence were read using a Varioskan^®^ Flash (Thermo Scientific, Shanghai, China) reader, respectively. The promoter activity of each plasmid was evaluated by normalizing the average of firefly luciferase activity to the ratio of Renilla luciferase activity. All the data were obtained from three independent transfection experiments performed in triplicate.

## 5. Conclusions

In this paper, *HdHSP70* was inducible by pathogenic infection, thermal stress, and combined stress, thus indicating that *Hd*HSP70 is involved in the regulation of innate immune as well as the stress response. The promotor structure and transcriptional analysis suggested that NF-1 and NF-κB may be two important transcription factors, which regulate the expression of *HdHSP70* gene.

## Figures and Tables

**Figure 1 molecules-24-00162-f001:**
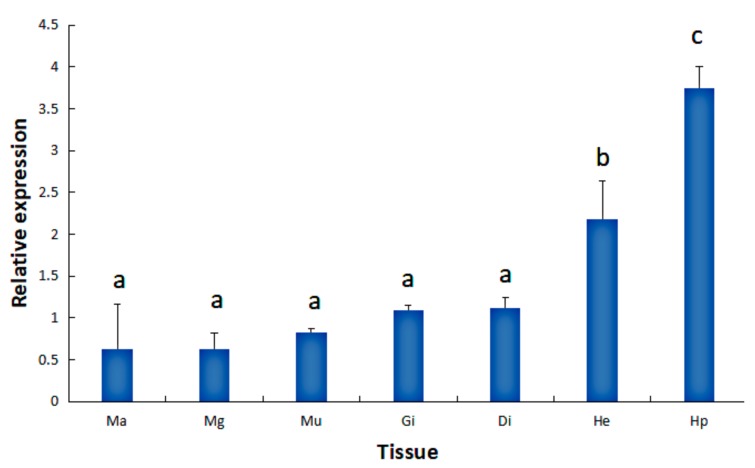
Expression levels of *HdHSP70* in different tissues of *H. diversicolor.* Tissues are mantle (Ma), mucous gland (Mg), muscle (Mu), gills (Gi), digestive tract (Di), hemocytes (He), and hepatopancreas (Hp). *β-actin* was selected as a reference gene. Values are means ± SD of biological replicates (*n* = 6). The different letters on the error bars represent significant difference, *p* < 0.05.

**Figure 2 molecules-24-00162-f002:**
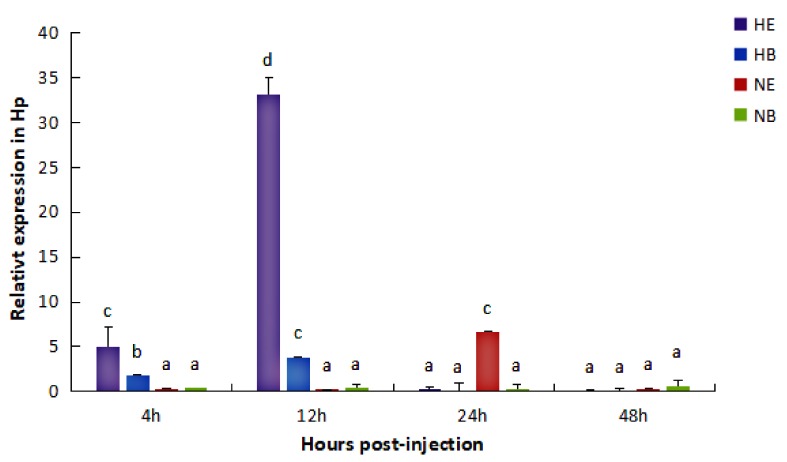
Expression level of *HdHSP70* in Hp was detected by qRT-PCR. Abalones were maintained for 96 h at increased temperature (30 °C) as thermal stress condition, after 96 h they were injected with 50 μL of live *V. parahaemolyticus* (1.1 × 10^8^ cfu/mL) (HE) or injected with 50 μL of 0.9% NaCl (HB); Abalones were maintained for 96 h at normal temperature (25 °C), after 96 h they were injected with 50 μL of live *V. parahaemolyticus* (1.1 × 10^8^ cfu/mL) (NE) or injected with 50 μL of 0.9% NaCl (NB); Samples were collected at 4, 12, 24 and 48 h after injection. *β-actin* was selected as a reference gene. Values are means ± SD of biological replicates (*n* = 6). The different letters (a, b, c, and d) on the error bars represent significant difference (*p <* 0.05) between challenged and control group. HE: thermal stress and injected *V. parahaemolyticus*, HB: thermal stress and injected NaCl. NE: normal condition and injected *V. parahaemolyticus*, NB: normal condition and injected NaCl. Hp: hepatopancreas.

**Figure 3 molecules-24-00162-f003:**
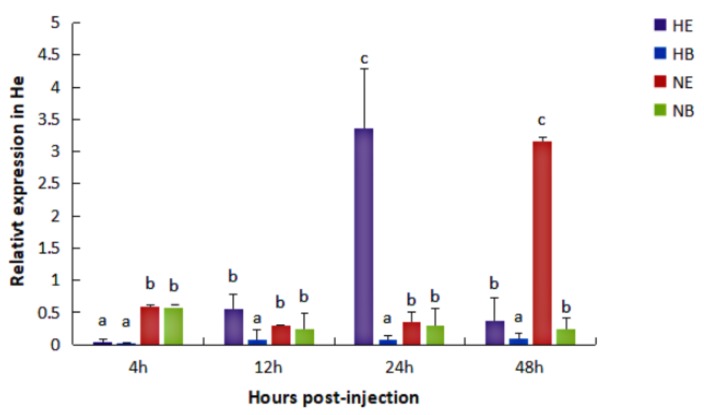
Expression level of *HdHSP70* in He was detected by qRT-PCR. Abalones were maintained for 96 h at increased temperature (30 °C) as thermal stress condition, after 96 h they were injected with 50 μL of live *V. parahaemolyticus* (1.1 × 10^8^ cfu/mL) (HE) or injected with 50 μL of 0.9% NaCl (HB); Abalones were maintained for 96 h at normal temperature (25 °C), after 96 h they were injected with 50 μL of live *V. parahaemolyticus* (1.1 × 10^8^ cfu/mL) (NE) or injected with 50 μL of 0.9% NaCl (NB); Samples were collected at 4, 12, 24 and 48 h after injection. *β-actin* was selected as a reference gene. Values are means ± SD of biological replicates (*n* = 6). The different letters (a, b, and c) on the error bars represent significant difference (*p* < 0.05) between challenged and control group. HE: thermal stress and injected *V. parahaemolyticus*, HB: thermal stress and injected NaCl. NE: normal condition and injected *V. parahaemolyticus*, NB: normal condition and injected NaCl. He: hemocytes.

**Figure 4 molecules-24-00162-f004:**
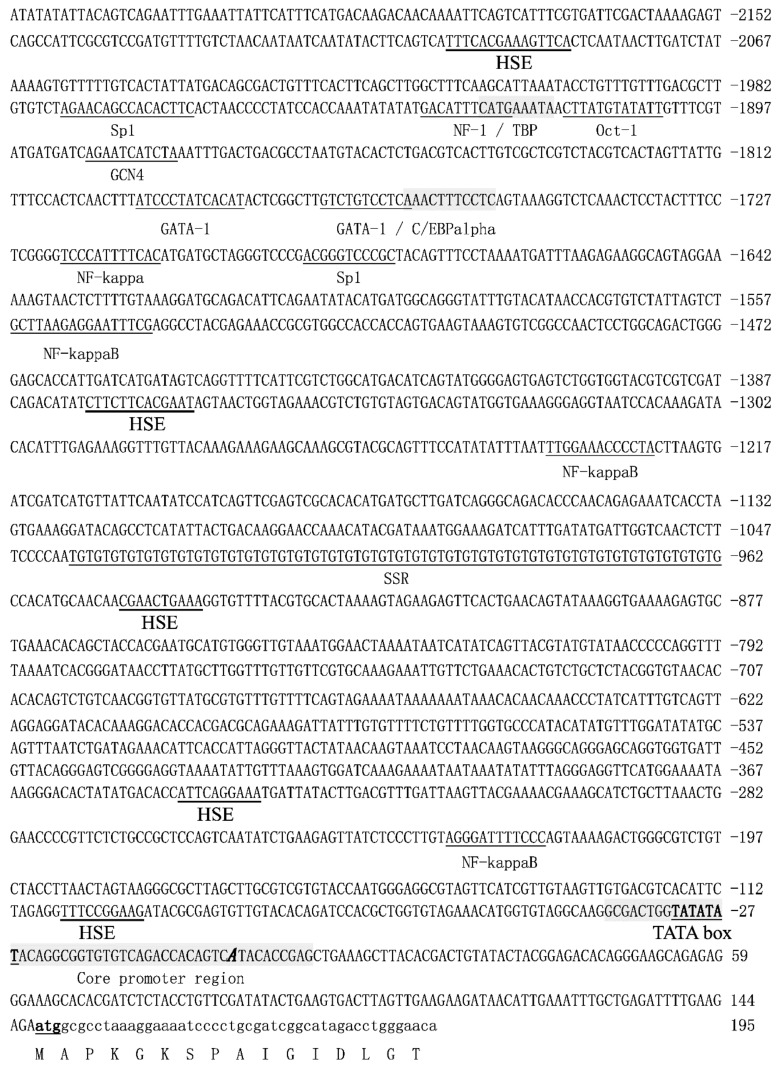
The nucleotide sequence of the 5′-flanking region of HSP70. The potential binding site of the transcription factors were marked with short thin line. Overlapping binding sites are indicated by shading. The predicted core promoter region is shaded, the transcription start site in bold letter, and is located at 1, and the translation start site (atg) in bolded and underlined. The heat-shock element HSE is marked with a short thick line and SSR with a long thin line.

**Figure 5 molecules-24-00162-f005:**
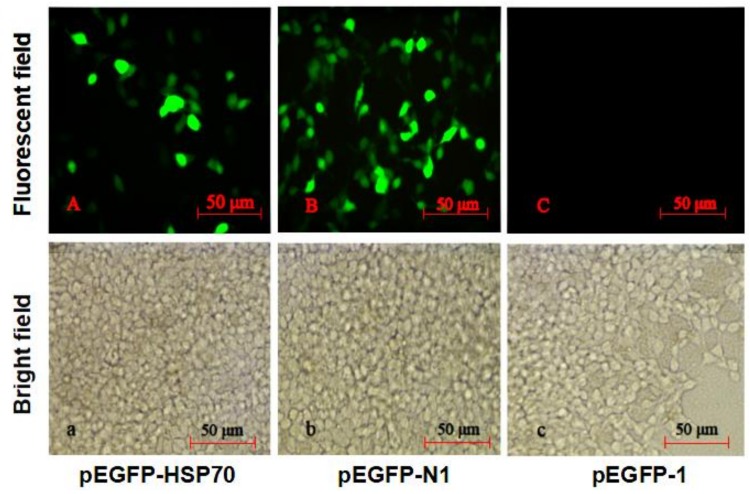
The expression of pEGFP-HSP70 in HEK293FT cells. HEK293FT cells were transfected with pEGFP-HSP70 vector that used the *HdHSP70* target promoter (**A**), pEGFP-N1 vector served as a positive control (**B**), and pEGFP-1 vector used as a negative control (**C**). After transfected for 24 h, the green fluorescence (EGFP) can be detected in A and B (target plasmid and positive control), but not in **C** (negative control) under a fluorescence microscope. Fluorescent fields are shown in (**A**, **B** and **C**) and bright fields are observed in (**a**, **b**, and **c**) separately. Values are means ± SD of biological replicates (*n* = 3). Scale Bars were 50 μm.

**Figure 6 molecules-24-00162-f006:**
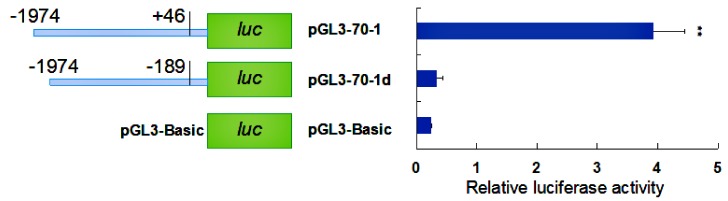
The relative activity of the *HdHSP70* gene with and without the predicted core promoter region. The plasmid with one from −1974 to +46 containing the core promoter region was named pGL3-70-1 and the other one from −1974 to −189 lacking the core promoter region was named pGL3-70-1d. Values are means ± SD of biological replicates (*n* = 3). The significant difference is indicated by a (**) at *p* < 0.01 as compared with the control (pGL3-Basic). Luc: luciferase expression plasmids.

**Figure 7 molecules-24-00162-f007:**
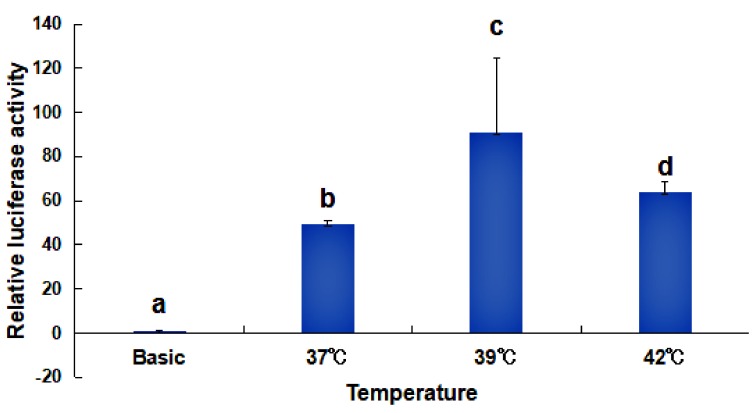
Changes of *HdHSP70* promoter activity in HEK293FT cells under high temperatures. The cells were exposed to high temperatures (37 °C, 39 °C and 42 °C) for 40 min. Values are means ± SD of biological replicates (n = 3). The different letters on the error bars indicate different significant differences, *p* < 0.05.

**Figure 8 molecules-24-00162-f008:**
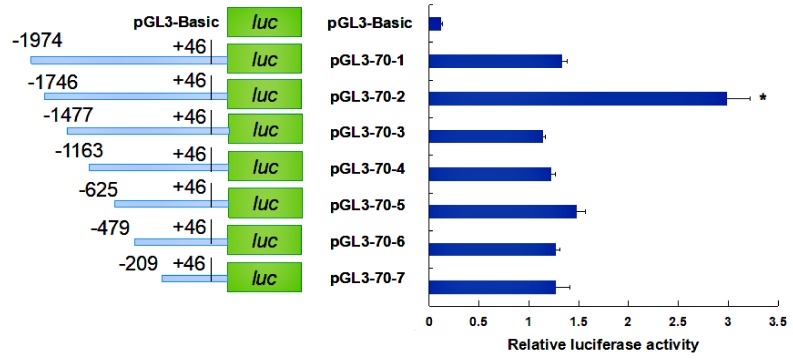
Activity analysis of *HdHSP70* gene promoter in HEK293FT cells. Based on the length of the seven fragments, the recombinant plasmids were named pGL3-70-1, pGL3-70-2, pGL3-70-3, pGL3-70-4, pGL3-70-5, pGL3-70-6 and pGL3-70-7. The pRL-TK vector containing Renilla luciferase gene was transfected as an internal reference to correct the transfection efficiency. The pGL3-Basic plasmid was served as a negative control. The significance of luciferase activity differences was analyzed using one-way ANOVA (Analysis of Variance) test. The values were averaged from three independent replicates. Values are means ± SD of biological replicates (*n* = 3). The significant difference between pGL3-70-2 and pGL3-70-1 or pGL3-70-3 is indicated by a (*) at *p* < 0.05. Luc: luciferase expression plasmids.

**Figure 9 molecules-24-00162-f009:**
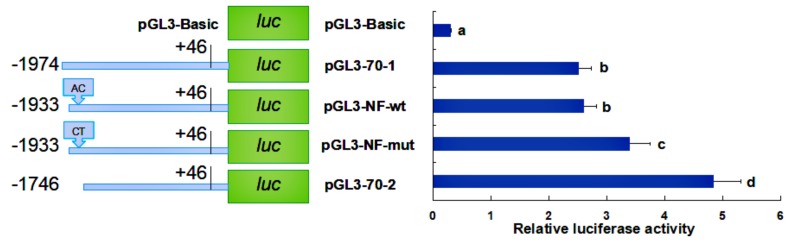
Activity analysis of the site-directed mutation plasmid pGL3-NF-mut. The distance between pGL3-70-1 and pGL3-70-2 contains a sole NF-1 binding site (TGACATTTCA) in the promoter of *HdHSP70*. In addition, the binding sites of the transcription factor NF-1 is TGACATTTCA, so the recombinant plasmid was named pGL3-NF-wt. When AC was mutated into CT, the sequence became TGCTATTTCA, and the mutant recombinant plasmid was named pGL3-NF-mut. The pGL3-Basic plasmid was served as a negative control. Values are means ± SD of biological replicates (*n* = 3). The different letters on the error bars represent significant differences, *p* < 0.05. Luc: luciferase expression plasmids.

**Figure 10 molecules-24-00162-f010:**
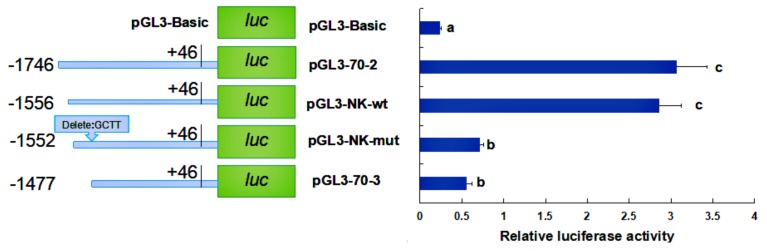
Activity analysis of the site-deleted mutation plasmid pGL3-NK-mut. The distance between pGL3-70-2 and pGL3-70-3 contains a sole NF-κB binding site in the promoter of *HdHSP70*. In addition, the binding sites of the transcription factor NF-κB is GCTTAAGAGGAATTT, and the recombinant plasmid was named pGL3-NK-wt. When GCTT was deleted, the new sequence became AAGAGGAATTT, and the mutant recombinant plasmid was named pGL3-NK-mut. The pGL3-Basic plasmid was served as a negative control. Values are means ± SD of biological replicates (*n* = 3). The different letters on the error bars represent significant differences, *p* <0.05. Luc: luciferase expression plasmids.

**Figure 11 molecules-24-00162-f011:**
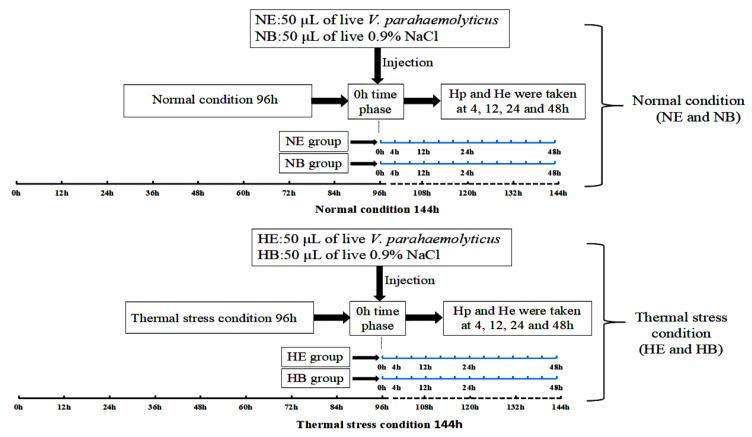
The frame chart of experiments in this study. Under the normal condition (25 °C), two groups of animals were maintained for 96 h. After 96 h, NE group were injected with 50 μL live *V. parahaemolyticus* (1.1 × 10^8^ cfu/mL in 0.9% NaCl), NB group were injected with 50 μL 0.9% NaCl. Under the thermal stress condition (30 °C), two groups of animals were maintained for 96 h. After 96 h, HE group were injected with 50 μL live *V. parahaemolyticus* (1.1 × 10^8^ cfu/mL in 0.9% NaCl), HB group were injected with 50 μL 0.9% NaCl. Hp and He were collected at time phases 4, 12, 24 and 48 h. Values are means ± SD of biological replicates (*n* = 6). HE: thermal stress and injected *V. parahaemolyticus*, HB: thermal stress and injected NaCl. NE: normal condition and injected *V. parahaemolyticus*, NB: normal condition and injected NaCl. Hp: hepatopancreas.

**Table 1 molecules-24-00162-t001:** Primers used in this study.

Primer Name	Start	End	Length (bp)	Primer Sequence (5′→3′)	Tm (°C)	Used for
HSP70-1				CTGAGATTTTGAAGAGAATGGC	56.2	Genome walking and Tail-PCR
HSP70-2				AGAGAGGGAAAGCACACGAT	56.0	
HSP70-3				TATGGTGAAAGGGAGGTAATCC	57.7	
HSP70-4				ATGGGGAGTGAGTCTGGTGG	59.1	
HSP70-5				CAACTCCTGGCAGACTGGGGA	64.5	
pGL3-70-r				CCGCTCGAGTGTGTCTCCGTAGTATACAGTCGT	56.0	
pGL3-70-1	−1974	46	2020	CGGGGTACCGAACAGCCACACTTCACTAACC	56.5	*HSP70* promoter activity
pGL3-70-2	−1746	46	1792	CGGGGTACCGTCTCAAACTCCTACTTTCCTCG	57.1	
pGL3-70-3	−1477	46	1523	CGGGGTACCTGGGGAGTGAGTCTGGTGGT	59.5	
pGL3-70-4	−1163	46	1209	CGGGGTACCTCAGGGCAGACACCCAACAG	61.3	
pGL3-70-5	−625	46	671	CGGGGTACCAGTTAGGAGGATACACAAAGGACA	57.1	
pGL3-70-6	−479	46	525	CGGGGTACCAAGTAAGGGCAGGGAGCAGG	60.9	
pGL3-70-7	−209	46	255	CGGGGTACCACTGGGCGTCTGTCTACCTT	56.2	
pGL3-70-1dr				CCGCTCGAGAAGGTAGACAGACGCCCAGT	56.1	
P-rt-f				CATAGACGAGGGCTCCATGT	57.3	qRT-PCR
P-rt-r				TCATGGCTCGTGTGTTGTTG	58.1	
β-actin-f				CCGTGACCTTACAGACTACCT	53.6	
β-actin-r				TACCAGCGGATTCCATAC	54.2	

Notes: HSP70-1 to HSP70-5 was used as primer for genome walking and Tail-PCR amplification. pGL3-70-r is the shared reverse primer, and pGL3-70-1 to pGL3-70-7 represent the forward primers for PCR fragments with different length. pGL3-70-1dr is the reverse primer for amplifying the fragment from −1974 to −189 bp without the core promoter region. P-rt-f and P-rt-r are forward and reverse primers for qRT-PCR, and β-actin-f and β-actin-r are internal reference primers for qRT-PCR. Xho I and Kpn I restriction sites are labeled with short thin lines and protective bases are CGG/CCG).

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
