# Peer review of "Responses of HSP70 Gene to Vibrio parahaemolyticus Infection and Thermal Stress and Its Transcriptional Regulation Analysis in Haliotis diversicolor"

_molecules, 2019, doi:10.3390/molecules24010162_

Reviewer 1 Report

Comments to Fang et al.,

The manuscript entitled “Responses of HSP70 gene to Vibrio parahaemolyticus infection and thermal stress and its transcriptional regulation analysis in Haliotis diversicolo” is a very interesting study.

In this study, the authors aimed at characterizing HSP70 gene in abalone and to determine the immunological effects of the hdhsp70 gene upregulation (by temperature/bacterial infection) on the innate immune response. However, there were various points especially in the results and discussion sections which need major amendment/clarifications. Without a proper restructuring of the manuscript, I am afraid, the manuscript can be considered for publication. My comments are listed below:

What is very confusing for me in this study was the methodology. The experimental design and the purpose of the study are not well defined.

There are many grammatical errors in the manuscript and the authors need to pay attention to them. I addressed some but the authors must thoroughly revise the manuscript for grammatical errors - especially in the results and discussion sections.

Abstract

The objectives of the study and the commercial importance of the species must be addressed in this part.

L14. The results showed that HdHSP70 gene was ubiquitously expressed in sampled tissues and was the highest in hepatopancreas, followed by haemocytes.

L17. the HdHSP70 gene was significantly up-regulated by Vibrio parahaemolyticus infection.  Please refer to the tissue or the sample.

L17. combined stress: Needs to be explained.

L17. Indicating that

L18. innate immunity

  Introduction

 Please follow the correct format for referencing in the text.

The introduction needs to be structured with more links between various parts. As of now, it appears to contain various independent information. Also, the objectives of the study, the importance of the selected pathogen and the importance of HSP70 for preventing Vibriosis (in general) in invertebrates must be addressed.  There are several studies on this. Also, more information on the TFs that were studied in this work and their links with HSP70 need to be provided.

L39. affecting the growth and health of

L40. In a study it was shown that, when the temperature was elevated from 28 to 32 °C, the phenoloxidase and phagocytic activities were negatively affected in the animals, indicating the importance of the environmental factors, mainly temperature, on the innate immune system regulation of the abalones.

L43. Similarly, Vibrio parahaemolyticus has been described as the pathogen which significantly affects the culture of H. diversicolor by invading the innate immune system of the cultured animals.

L44. Pathogenic bacteria infections  can easily lead to the abrupt death of abalone and cause catastrophic losses to aquaculture farmers – Delete

L45: Therefore, understanding the mechanism behind the adverse effects of environmental factors, mainly elevated temperature and pathogen invasion, on the innate immune system of abalone is of high importance.

L47. The link between innate immunity and molecular chaperones need to be made. For this, You can refer to the following publications:

1)       Reactive oxygen species generated by a heat shock protein (HSP) inducing product contributes to Hsp70 production and Hsp70-mediated protective immunity in Artemia franciscana against pathogenic vibrios

2)       Non-Lethal Heat Shock Increased Hsp70 and Immune Protein Transcripts but Not Vibrio Tolerance in the White-Leg Shrimp

L50. Heat shock proteins (HSPs) are

L52. Significant regulation of HSPs is a critical part of the heat shock response and primarily guided

53 by heat shock factors – Please rephrase

L56. Therefore, the function  and regulation mechanism of HSPs always draw keen interest from researchers -rephrase

L76. under various stress conditions – Delete

L87. and during early embryonic development

L97. reported that

 Results

Please provide the full form of your abbreviations when they are used for the first time.  Example: HP and HE. Please also maintain uniformity in using them.

Figure captions need to be explained with more details. If applicable, please mention the number of replicates, dose of infection (CFU cells/ml), number of cells, bioinformatics software that were used or imaging devices. Revise the figures titles as well and make them more clear and detailed.

Tables may help in simplifying the obtained results. Please make a table with abbreviations and time points of sampling as well as the time that the results showed significant values. Similar tables can be made for the mutations and luciferase activity.

L113. The expression of constitutive HdHSP70 gene was studied in different tissues of healthy abalone

L113. Please mention the name of the tissues

L114. with the significantly higher expression level being in Hp (P<0.05), followed by He (P <0.05).< p="">

L115. The gene was also expressed in other examined tissues, but no significant difference among different tissues was found – Delete

L116. Based on the obtained results, Hp and He were set as experimental tissues in different

stress groups.

Fig. 2. Figure caption needs more explanation, including number of replicates and type of analysis. Mainly the internal control based on which the relative analysis was performed should be mentioned. Also, the tissue from which the sampling was performed in abalone groups (NE and NB) must be mentioned. Please adapt all the figure captions accordingly.

L138. In the He, under normal condition, the expression level of HdHSP70 was significantly higher (P

 <0.05) in NE group than that in NB group at 48h, but NE group and NB group were not different at 4 h, 12 h and 24 h. Under the thermal stress condition, the expression level of HdHSP70 in HE group was significantly higher (P <0.05) than that in HB group at 12 h, 24 h and 48 h, but HE group, and HB group were not different at 4 h (Fig. 3). This part requires more explanation of the samples and treatments for a better read. Also requires a rephrase and minor grammatical changes.

L156. Promoter, starting from -40 to +10 bps region

L167. translation start site in bold letters

L168. The heat shock element HSE is marked with a short line, for SSR Long line under the line – not clear

L181. A  brief overview of luciferase importance for your study and its link to HSP70 is required.

Fig. 6. Analysis of the relative activity of the HdHSP70 gene with and without the predicted core promoter region

L192. After exposing HEK293FT cells to high temperatures, ranging from 37 to 39 °C and also 42 °C for 40 mins, we found a significant increase in the activities of luciferase at 39 °C and 42 °C compared to 37 °C (P<0.05), with the highest activity being observed at 39 °C (Fig. 7).

L199. The results showed that, compared to the control group (pGL3-Basic), the luciferase activities had an increasing tendency from pGL3-70-1 to pGL3-70-2, and pGL3-70-2 promoter activity reached the highest level, followed by a significant decrease in pGL3-70-3 activity, but there was no significant difference between pGL3-70-3 to pGL3- 70-7 (P<0.05; Fig. 8) – rephrase

L216. the sequence of NF-1 binding site was mutated to CT, and the sequence 216 became TGCTATTTCA instead of …. The mutant recombinant plasmid was named as pGL3-NF-mut.

L217. A recombinant plasmid containing wild

L218. The luciferase activity analysis showed that in pGL3-NF-mut, luciferase activity was significantly increased compared to the same in pGL3-NF-wt  (P<0.05; Fig. 9).

L220. Similarly, the distance between the two deletion fragments of pGL3-70-2 to pGL3- 70-3 contains a sole NF-κB binding site in the promoter of HdHSP70 – not clear, Please rephrase and provide more explanation.

L221. A recombinant plasmid containing

L223. recombinant plasmid in which GCTT was deleted, to achieve a sequence AAGAGGAATTT, was constructed and named as pGL3-NK-mut - rephrase

Fig 9 and 10. Please rephrase the caption and provide more details for a better understanding of the reader.

Discussion

The discussion contains very scattered information. Please start with the importance of HSP70 for animal’s tolerance against environmental (heat stress/pathogen) factors. In this regard, there are several studies in Arabidopsis, Drosophila, Artemia, Daphnia, and nematode worm.

In your studies, you didn’t score the survival of your animals and cells after the heat treatment and/infection. Therefore, based on your gene expression values, it is very difficult to draw a conclusion for a positive immune response at the phenotypic level. Please avoid this. Please avoid repetition of your results in your discussion.

 L 239. It has been well known that the complex environmental factors including changes in temperature, hypoxia, and infection with various species of pathogens can affect physiological activities of aquatic animals.

L241. However, the influence of environmental factors on the immune

242 function of molluscs has not received sufficient attention yet - delete

L242 – 247. Rephrase.

L255. Many studies have similar results, for example, the higher expression of the HSC70 mRNA in HP and HE of H. diversicolor [43], and higher expression of HSP70 mRNA in the liver of chickens [44] and in the blood of Oncorhynchus mykiss [45] – Rephrase; also there are many studies performed in more closely related (invertebrate) species.

L272. Regardless of the normal or thermal stress condition in the Hp and He, the expression level of HdHSP70 in V. parahaemolyticus infection group (NE and HE group) was significantly higher than the control group at some time phases – Rephrase

 Materials and methods:

 The part related to the infection model is not well explained.

Reviewer 2 Report

The manuscript “Responses of HSP70 gene to Vibrio parahaemolyticus infection and thermal stress and its transcriptional regulation analysis in Haliotis diversicolor” submitted to Molecules has been reviewed. It worked on the HSP70 in abalone. The gene expression profiling, its promoter structure and activity were investigated.

The experiment was designed in logic, and data collected supported the conclusion and research objectives. However, the data interpretation needs an elaboration, particularly expression profiling is a simple and partial description. The overall English writing is about average and polishing and proofreading are needed before next submission. Please see comments in attachment.
